# Influence of Hyponatremia on Spinal Bone Quality and Fractures Due to Low-Energy Trauma

**DOI:** 10.3390/medicina57111224

**Published:** 2021-11-10

**Authors:** Katharina Jäckle, Friederike Klockner, Daniel Bernd Hoffmann, Paul Jonathan Roch, Maximilian Reinhold, Wolfgang Lehmann, Lukas Weiser

**Affiliations:** Department of Trauma Surgery, Orthopaedics and Plastic Surgery, University Medical Center Göttingen, Robert-Koch Str. 40, 37075 Göttingen, Germany; katharina.jaeckle@med.uni-goettingen.de (K.J.); friederike.klockner@med.uni-goettingen.de (F.K.); daniel.hoffmann@med.uni-goettingen.de (D.B.H.); jonathan.roch@med.uni-goettingen.de (P.J.R.); maximilian.reinhold@med.uni-goettingen.de (M.R.); Wolfgang.Lehmann@med.uni-goettingen.de (W.L.)

**Keywords:** osteoporosis, chronic hyponatremia, QCT, BMD, spine

## Abstract

*Background and Objectives:* Hyponatremia is the most common electrolyte disorder in elderly and associated with increased risk of falls. Clinical studies as well as small animal experiments suggested an association between chronic hyponatremia and osteoporosis. Furthermore, it has been assumed that subtle hyponatremia may be an independent fracture risk in the elderly. Therefore, this study was designed to evaluate the possible influence of chronic hyponatremia on osteoporosis and low-energy fractures of the spine. *Materials and Methods:* 144 patients with a vertebral body fracture (mean age: 69.15 ± 16.08; 73 females and 71 males) due to low-energy trauma were treated in a level one trauma center within one year and were included in the study. Chronic hyponatremia was defined as serum sodium < 135 mmol/L at admission. Bone mineral density (BMD) of the spine was measured using quantitative computed tomography in each patient. *Results:* Overall, 19.44% (n = 28) of patients in the low-energy trauma group had hyponatremia. In the group with fractures caused by low-energy trauma, the proportion of hyponatremia of patients older than 65 years was significantly increased as compared to younger patients (*p*** = 0.0016). Furthermore, there was no significant gender difference in the hyponatremia group. Of 28 patients with chronic hyponatremia, all patients had decreased bone quality. Four patients showed osteopenia and the other 24 patients even showed osteoporosis. In the low-energy trauma group, the BMD correlated significantly with serum sodium (r = 0.396; *p**** < 0.001). *Conclusions:* The results suggest that chronic hyponatremia affects bone quality. Patients with chronic hyponatremia have an increased prevalence of fractures after low-energy trauma due to a decreased bone quality. Therefore, physicians from different specialties should focus on the treatment of chronic hyponatremia to reduce the fracture rate after low-energy trauma, particularly with elderly patients.

## 1. Introduction

Osteoporosis is a disease of the skeletal system that affects both sexes. Despite the by now promising treatment options, osteoporosis still represents a challenge for medicine. Osteoporosis can be divided into different forms. The most prevalent form is postmenopausal osteoporosis, which results from estrogen deficiency [1]. Osteoporosis is associated with an increased risk of bone fractures, usually due to impaired remodeling of bone substance and the corresponding pathological bone microarchitecture. Diagnosis of osteoporosis is usually very late, since the disease is clinically silent until the first fracture occurs. Due to the demographic development, several studies have predicted an enormous increase in osteoporosis and depending fractures in coming decades [2,3]. Diagnostic and therapeutic measures for, in particular, postmenopausal osteoporosis have therefore become an increasingly important topic of both preclinical and clinical research.

Osteoporosis primarily affects the trabecular bones, such as the proximal femur, the distal radius, or the vertebral bodies. In addition to genetic-, hormonal-, or medication-induced factors, which include glucocorticoids, additional factors, such as alcohol, nicotine, and endocrine factors, are known to influence the bone disease [4,5,6,7].

In addition to osteoporosis, hyponatremia is the most common electrolyte disorder in the elderly [8]. It is defined by sodium levels below 135 mmol/L in the serum [9]. Chronic hyponatremia has been shown to be associated with an increased risk of falls, attention deficit, and agitation [10]. In small animal experiments, chronic hyponatremia resulted in a significant reduction of the bone mass in approximately one-third of the cases [11], resulting in a deterioration of both trabecular and cortical bone properties [11]. Additionally, several clinical studies have suggested an association between chronic hyponatremia and osteoporosis, and a significant prevalence of hyponatremia in elderly patients with low-energy trauma has been shown [8,11,12,13,14]. Moreover, a more recent study suggested that subtle hyponatremia may be an independent fracture risk in the elderly [15].

A retrospective cohort study was performed to evaluate the prevalence and impact of chronic hyponatremia on spinal bone quality by quantitative computed tomography (QCT) analysis. The study included patients with vertebral body fractures in the lumbar spine due to a low-energy trauma, asking whether osteoporosis is associated with chronic hyponatremia in these patients. Furthermore, the serum sodium levels of the patients were examined and compared with patients having vertebral fractures due to a high energy trauma.

## 2. Materials and Methods

### 2.1. The Patient Collective for Study

The present study was approved by the ethics committee of the University Medical Center (approval number: AN 2/6/21) in compliance with the Helsinki Declaration.

The inclusion criteria for the reported retrospective cohort study were defined by a patient collective with vertebral body fractures at the lumbar spine due to a low-energy trauma and osteoporosis or osteopenia detected by quantitative computed tomography (QCT), who were treated at a university level in the trauma center during a one-year period (2017–2018). Low-energy trauma was defined as falls from standing height or less than 1 m and fractures without trauma. As a control group, a patient population with fractures after high-energy trauma who received a CT diagnostic (“control group”) were selected.

The serum sodium level on hospital admission of the patient was the major determining factor for the classification of hyponatremia. In this study, chronic hyponatremia was defined as a serum sodium level below 135 mmol/L on admission, and the level had to remain persistently low during the inpatient stay. Isolated low serum sodium levels were not considered as hyponatremia.

Based on these criteria, 199 patients (mean age: 65.39; 98 females and 101 males) with vertebral body fractures at the lumbar spine due to a low-energy trauma were identified. As controls, a total of 83 patients with high-energy bone fractures were included. All patients had received CT diagnostics. These CT data sets were used for quantitative computed tomography (QCT) analyses (QCT Pro^®^, version 6.1, Mindways Software; Kiel, Germany) in the lumbar spine region. The QCT analysis is a medical technique used to measure bone mineral density (BMD). It uses standard computed tomography (CT) diagnostics with a calibration standard to convert Hounsfield Units (HU) of the CT image to bone mineral density values [16]. Two independent measurements of BMD were made, and the mean values were calculated.

In 36 cases, the complete data set was not available. For this reason, these patients were excluded from the study. Furthermore, 19 patients were excluded due to cancer and cancer-related bone metastases. Thus, the study included 144 patients (mean age: 69.15 ± 16.08; 73 females and 71 males) (see Table 1). The control group was composed of 83 patients (mean age: 66.65 ± 15.60; 40 females and 43 males) with vertebral body fractures including high-energy trauma fractures of the spine (see above and Table 1). Exclusion criteria were again pathological fractures or tumor diseases.

### 2.2. Statistics

Statistical analysis was performed using the D’Agostino–Pearson test to check for normal distribution. The significance calculation was based on the Wilcoxon–Mann–Whitney test and the significance level was set to alpha = 5%. Furthermore, the Pearson correlation test was performed, and the confidence interval was set to 95%. For all statistical tests, the statistics software Graphpad Prism 8 (version 8.1.1 for mac) was used.

## 3. Results

### 3.1. Comparison between Patients with Low-Energy Trauma and the Control Group

On admission and during the inpatient stay, 19.44% (n = 28) of the patients from the low-energy trauma group had hyponatremia (see Figure 1). In this group, the mean serum sodium level was 132.61 mmol/L ± 3.39. The mean age of this patient population was 69.15 years and 50.69% (n = 73) were females.

The mean age and gender distribution were not significantly different from the 83 patients in the control group without hyponatremia (66.65 years, 48.19% female; *p* = 0.8695). Furthermore, there was no significant difference in body mass index (BMI) between the two groups of patients, i.e., the mean BMI value was 26.64 kg/m^2^ ± 4.71 in the chronic hyponatremia group compared to 28.39 kg/m^2^ ± 6.17 in the control group (*p* = 0.3639).

The low-energy patient cohort with chronic hyponatremia was divided into two age groups, i.e., younger and older than 65 years, respectively (see Figure 1a). In the group of patients younger than 65 years, 1.96% (1 of 51 patients) showed chronic hyponatremia. In the collective of patients who were older than 65 years, 29.03% (27 of 93 patients) were affected by hyponatremia. These differences between the two age groups were statistically significant (*p*** = 0.0016).

In the control group, all patients showed normal serum sodium levels on admission (*p*** = 0.0026). It should be noted, however, that a single serum sodium level value below 135 mmol/l during the course of hospitalization was found in one of the patients. This outlier was not considered as hyponatremia, since it reflects the normal range of fluctuation of blood values in healthy patients.

### 3.2. Hyponatremia Is Increased in the Low-Energy Trauma Group of Patients, but No Gender Difference Was Observed

In order to see whether there is a gender-specific difference between the two groups studied, i.e., the low-energy group and the control group, respectively, the distribution of males and females with hyponatremia within the two groups was compared. In the low-energy trauma group, the gender distributions of hyponatremia patients (female 50.69%, 73 of 144 patients) and non-hyponatremia patients (female 48.19%, 40 of 83 patients) were very similar (*p* = 0.1542).

In the age group below 65 years of the low-energy trauma patients, only one male showed hyponatremia (female: 0%, 0 of 18 patients; male: 3.03%, 1 of 33 patients). In the group of patients above 65 years, female patients showed a higher rate of hyponatremia (32.73%; n = 18) than male patients (23.68%; n = 9). However, this difference was not statistically significant (*p* = 0.8566) (Figure 1b). Thus, no significant gender difference could be noted with respect to patients affected by hyponatremia.

### 3.3. Distribution of Osteoporosis in Patients and Its Association with Hyponatremia

The bone mineral density in the low-energy and high-energy trauma patients was examined and compared to the standard values of the QCT Spinal Bone Mineral Density Classification of the American College of Radiology [17] (Table 2). In the low-energy trauma group, 17.36% of patients (n = 25) showed normal bone mineral density (above 120 mg/cm^3^, see Table 3; female: 4.86%; male: 12.50%). A reduced bone mineral density, i.e., osteopenia (80–120 mg/cm^3^), was observed with 50 patients (34.72%). Of those, 22 patients were female (15.28%) and 28 were male (19.44%). Moreover, 69 patients (47.92%) had an ever-lower bone mineral density < 80 mg/cm^3^ and thus suffered from osteoporosis. They included 43 females (29.86%) and 26 males (18.06%) (see Table 3, Figure 2). In the high-energy fracture patient control group, 29 patients (34.94%) showed normal bone mineral density (female: 16.87%; male: 18.07%). These values were notably higher as compared to the low-energy group, but again the difference was not statistically significant (*p* = 0.7568). Furthermore, osteopenia was also present in 30 (36.14%) of the patients (female: 15.66%; male: 20.48%). These differences are overall comparable with the results obtained with the low-energy group and were also not significant (*p* = 0.9077). Osteoporosis was diagnosed in 24 patients (28.92%) of the cases, with 13 female (15.66%) and 11 male patients (13.25%). The distribution of osteoporosis was therefore significantly lower in the control group than in the low-energy group (*p** = 0.0327), but no gender difference was observed.

None of the patients in the control group, i.e., high-energy trauma patients, showed hyponatremia. In the low-energy group, however, 28 patients (19.44%) showed chronic hyponatremia. Of these 28 patients, all patients were diagnosed with decreased bone quality as described above, i.e., four patients were diagnosed with osteopenia (female: n = 1; male: n = 3) and 24 patients with osteoporosis (female: n = 16; male: n = 8).

In summary, the data indicate a strong and statistically significant correlation of low-energy fractures and hyponatremia, and no hyponatremia patient was observed in the control group of patients with fractures due to a high-energy trauma.

### 3.4. Correlation between Bone Mineral Density (BMD) and Sodium Level

The correlation was calculated with the Pearson product-moment correlation coefficient between the bone mineral density (BMD) and the sodium level of the values from the patient collective. BMD and sodium level were positively correlated with a medium correlation for r = 0.396 in the low-energy group, which means that the higher the sodium value in the blood, the better the bone quality of the patient. The correlation between the two values was statistically significant with *p**** < 0.001 (see Figure 3).

## 4. Discussion

The study establishes that chronic hyponatremia is highly prevalent in patients with fractures due to a low-energy trauma. This correlation has been suggested by earlier studies [8,13,18,19,20] showing that elderly patients with hyponatremia are susceptible to fragility fractures. Chronic hyponatremia in about one-fifth (19.44%) of the patients who experienced a low-energy trauma was observed, while no hyponatremia patients were found within the high-energy trauma control group. The results therefore indicate that the incidence of low-energy fractures is strongly increased in hyponatremia patients, establishing a correlation between hyponatremia and low-energy fractures. This link is mediated via reduced bone quality to the level of osteopenia or even osteoporosis, implying an association rather than causality between fracture risk and hyponatremia. This causation must be noted in the absence of a multivariable regression analysis, which cannot be applied in a meaningful manner or would even be misleading, since the study does not involve the required large data sets for such an analysis. The results also exclude the possibility that the body mass index of the patients is relevant for low-energy traumas, since it was similar in the two groups of patients.

Previous studies reported a prevalence of hyponatremia in patients with fractures ranging between 9% and 13.6% [13,18,19,21]. As in the present study, they mainly focused on patients older than 60 years, but only one study [8] examined the prevalence of hyponatremia with respect to low-energy traumas. The data regarding the correlation of chronic hyponatremia, bone quality, and fractures in 19.44% of the cases are clearly above the maximum value reported in these earlier studies, and also higher than the value reported by Cumming et al. [8] (13.4% prevalence of hyponatremia on admission) who studied a cohort of older people with low-energy traumas (average 79 years; 78% females). In the present as well as in previous studies, a low sodium level (<135 mmol/L) on admission was the major determinant for the initial diagnosis of hyponatremia. To ensure a chronic hyponatremia diagnosis, the examination of sodium levels during the hospitalization of the patients was continued to focus the study on only those patients who showed persistently low sodium serum levels during the inpatient stay. Thus, patients were excluded with postoperative hyponatremia of short duration to ensure that the association of hyponatremia and low-energy traumas could be firmly established. This association is supported by the fact that no hyponatremia case was observed in the control group, i.e., bone fractures of patients with high-energy traumas.

It was observed that hyponatremia cases were significantly increased in the group of low-energy trauma patients who were older than 65 years (*p*** = 0.0016). This fact is not surprising, since younger patients also have fewer concomitant diseases and are thus generally healthier. It also suggests that hyponatremia increases with the age of patients [4,22].

The results also indicate that there was no significant difference with respect to the gender incidence of hyponatremia among the patients with or without a low-energy trauma. Even when the patients were divided into two age groups, i.e., patients who were younger and older than 65 years, respectively, the differences concerning the numbers of male and female hyponatremia patients were not significant. This finding supports the results of an earlier study showing that hyponatremia is equally distributed in the elderly male and female population without fractures [23].

The results also indicate a correlation between sodium levels and BMD, i.e., the higher the blood sodium level, the better the bone quality. The correlation between these two parameters was statistically significant (*p**** < 0.001). Although the correlation factor was low, with r = 0.396, a clear tendency concerning the correlation can be concluded, consistent with studies of animal experiments as reported by Verbalis et al. [11]. Using the rat model, the authors [11] reported that chronic hyponatremia is associated with a significant reduction in bone mass in approximately one-third of cases [11].

Notably, the presented study has some limitations, such as the uncertain duration of preexisting hyponatremia of the patients. Only hyponatremia on admission and continuous low sodium levels during hospitalization was diagnosed, without having data prior to the admission. Very acute or severe hyponatremia may present with serious complications, such as non-cardiogenic pulmonary oedema, hyponatremic encephalopathy, and neurological symptoms [4,22]. Such severe complications in the current study groups were not found, suggesting that most, if not all, of the patients included in the study belong to the 75–80% of the cases of hyponatremia that are mild and chronic [22]. These cases are considered asymptomatic despite being strongly associated with major geriatric conditions and multi-organ pathological changes [10,22,24]. Such conditions were not considered in the study since the first sodium measurement was taken only on the day of admission and followed up only during the usually short hospital stay. Nevertheless, the results of the present study are in line with earlier reports, suggesting that chronic hyponatremia is an important determinant of fracture risk [15,20].

Another limitation of the study is that the influence of medications on sodium levels in the patient population was not considered as has been investigated in several previous studies [8,13,18,25,26,27,28]. These studies reported, for example, that aldosterone antagonists and sartans have a significant impact on hyponatremia of patients [29] as has also been observed with selective serotonin re-uptake inhibitors (SSRIs) and thiazide, all of which induce hyponatremia [30,31,32,33,34]. This aspect concerning the influence of possible medication on hyponatremia and its subsequent effect on low-energy trauma fractures is not reflected in the results.

Although there are already several studies suggesting that chronic hyponatremia represents a risk factor for bone fractures due to low-energy traumas, this study is the first to examine the bone quality of hyponatremia patients by QCT analysis. QCT analysis is currently predominantly a research tool, but it has some important advantages over dual-energy X-ray absortiometry (DXA) in studies of the skeleton and is being increasingly used to determine bone quality [16]. In the low-energy group of trauma patients, 28 patients (19.44%) had chronic hyponatremia, which was associated in all cases with a significant decrease in bone quality. Four of these patients were affected by osteopenia and the other 24 patients by osteoporosis. These results make it clear that chronic hyponatremia is associated with decreased bone quality, confirming the earlier observation of Verbalis et al. [11]. This finding therefore provides a link between low-energy trauma fractures and hyponatremia via reduced bone quality and the level of osteopenia or even osteoporosis, implying an indirect causality between the fracture risk and hyponatremia.

## 5. Conclusions

The present study confirms a high prevalence of chronic hyponatremia in patients with fractures after low-energy trauma. It also highlights that chronic hyponatremia is a risk factor for low-energy fractures due to decreased bone quality combined with an increased risk of falls, attention deficit, and restlessness of elderly people. The last three points have been demonstrated in a previous study [10]. Furthermore, a direct correlation between serum sodium level and bone mineral density was shown. Accordingly, increasing attention must be given to the underestimated effects of chronic hyponatremia on bone fractures and elderly people with low sodium levels require targeted therapy as a precaution to reduce the risk of fractures due to low-energy trauma.

Taken together, the results suggests that chronic hyponatremia impairs bone quality and that chronic hyponatremia results in an increased prevalence of fractures after low-energy trauma due to decreased bone quality.

## Figures and Tables

**Figure 1 medicina-57-01224-f001:**
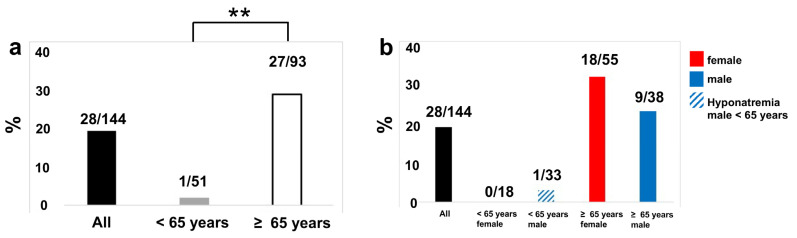
Distribution of hyponatremia. (**a**) Hyponatremia distribution in different age groups of the patient collective (n = 144); black bars represent the number of the whole population (n = 144), gray bars indicate hyponatremia in the age cohort younger than 65 years (n = 51), and white bars show the distribution of hyponatremia in the age cohort ≥ 65 years (n = 93; *p*** = 0.0016). Distribution of patients in percent shown on the y-axis. Age cohort of patients shown on the x-axis. (**b**) Gender distribution of hyponatremia among the two age groups, i.e., patients who are younger than 65 years and 65 years or older, respectively. The percentage of patients with hyponatremia is indicated on the y-axis, and the age groups on the x-axis.

**Figure 2 medicina-57-01224-f002:**
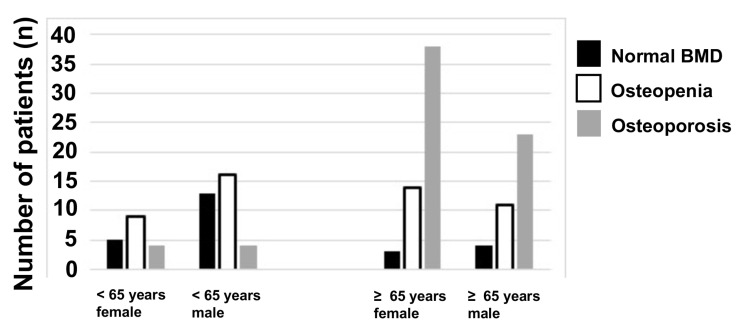
Distribution of bone mineral density (BMD) between the two age cohorts younger than 65 years and ≥65 years. Black bars show normal BMD, white bars show osteopenia, and grey bars indicate osteoporosis. Number of patients is indicated on the y-axis (n) and the different age groups (i.e., patients who are younger than 65 years and 65 years or older, respectively) are presented on the x-axis.

**Figure 3 medicina-57-01224-f003:**
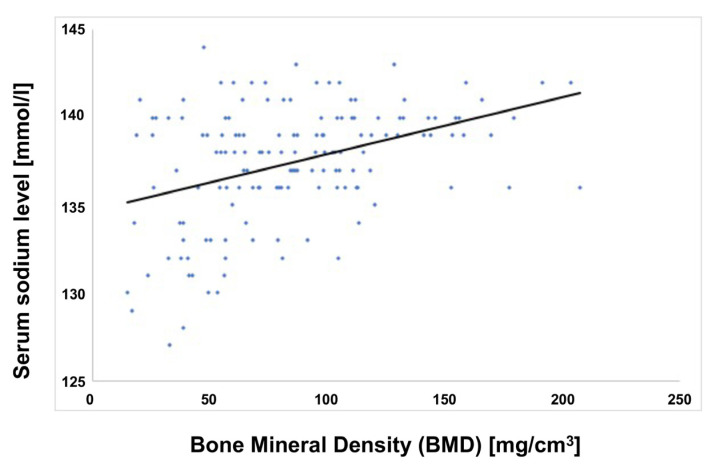
Correlation between bone mineral density (BMD) and serum sodium level. A positive correlation between BMD and serum sodium level (r = 0.396; *p**** < 0.001)) in the low-energy group (n = 144) is shown. The serum sodium levels [mmol/L] are indicated on the y-axis and the bone mineral density [mg/cm^3^] is presented on the x-axis.

**Table 1 medicina-57-01224-t001:** Baseline characteristics of the populations.

Description	Low-Energy Group (n = 144)	Control Group (n = 83)
Number of patients	144	83
Age range [years]	23–94	23–91
Age mean [years] ± SD	69.15 ± 16.08	66.65 ± 15.60
**Gender**		
Female [n]	73	40
Male [n]	71	43

**Table 2 medicina-57-01224-t002:** QCT Spinal Bone Mineral Density Classification of the American College of Radiology.

Bone Mineral Density Area	WHO-Category
>120 mg/cm^3^	Normal
80–120 mg/cm^3^	Osteoporosis
<80 mg/cm^3^	Osteopenia

With kind permission of American College of Radiology (ACR); (see: ACR QCT Spinal Bone Density Classification values. Available online: https://www.acr.org/-/media/ACR/Files/Practice-Parameters/qct.pdf; accessed on 10 September 2021).

**Table 3 medicina-57-01224-t003:** QCT Spinal Bone Mineral Density Classification of the American College of Radiology.

Description	Low-Energy Group (n = 144)	Control Group (n = 83)	*p*-Value
Osteoporosis [mg/cm^3^]	69 (♀: n = 43; ♂: n = 26)	24 (♀: n = 13; ♂: n = 11)	0.0327 (*)
Osteopenia [mg/cm^3^]	50 (♀: n = 22; ♂: n = 28)	30 (♀: n = 13; ♂: n = 17)	0.9077 (n.s.)
Normal BMD [mg/cm^3^]	25 (♀: n = 7; ♂: n = 18)	29 (♀: n = 14; ♂: n = 15)	0.7568 (n.s.)
**Hyponatremia**	28	0	0.0026 (**)
< 65 female [n]	0	0	-
< 65 male [n]	1	0	-
≥ 65 female [n]	18	0	0.8182 (n.s.)
≥ 65 male [n]	9	0	0.8182 (n.s.)

♀ = female; ♂ = male; n.s. = not significant; * = statistically significant; ** = statistically highly significant.

## Data Availability

The data that support the findings of this study are available on request from the corresponding author. The data are not publicly available due to privacy or ethical restrictions.

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
