# Peer review of "Influence of Hyponatremia on Spinal Bone Quality and Fractures Due to Low-Energy Trauma"

_medicina, 2021, doi:10.3390/medicina57111224_

Round 1

Reviewer 1 Report

As a general comment, this paper presents a research of real interest, even though based on known knowledge.

Title is short and concise.

Abstract, even though a little bit over 250 words, describes what was done, results and conclusions; kew words are clearly defined. Introduction provides necessary background and hypothesis for this retrospective study.

Material and methods: methods are accurate described, with inclusion and exclusion criteria well defined (osteoporosis or osteopenia, trauma level, trauma analysis, hyponatremia), patient cohort adequately reported, in terms of size, age and sex distribution.

Results section match methods section, being also adequately reported, by all correlations found: comparison between low-energy trauma and control group, hyponatremia in low energy trauma group, osteoporosis distribution and hyponatremia and finally, correlation between bone density and natremia.

Discussions starts with papers’ findings, presented in context of previous studies; limitations of this study are stated, but also the strength of this study: QCT analysis  for examining bone quality in hyponatremic patients, with obvious clinical relevance.

Conclusions are short, based on manuscript contained data.

Tables are clear, easy to understand.

References are up to date and correctly formatted.

Author Response

Medicina-1442547
Resubmission of our revised manuscript “Influence of hyponatremia on spinal bone quality and fractures due to low-energy trauma“ by Jäckle et al.

Dear Editors,
Thank you very much for providing a rapid and thorough reviewing process. One of the reviewers praised our work and did not make any suggestions for improvement. The second reviewer has some positive criticism and valuable suggestions which we address in our point-by-point response below. Overall, we are very thankful for the comments and for having the opportunity to resubmit our manuscript, following the advice/suggestions of the reviewer 2 which helped to improve our revised manuscript. 

As you will see, our point-by-point response explains how we addressed each comment of reviewer 2. The changes made in the revised manuscript are highlighted in green.

Kind regards,

K. Jäckle

Point-by-point response to the reviewer´s critique, comments and suggestions.

Reviewer #1: 
Comments and Suggestions for Authors
As a general comment, this paper presents a research of real interest, even though based on known knowledge.
Title is short and concise.
Abstract, even though a little bit over 250 words, describes what was done, results and conclusions; kew words are clearly defined. Introduction provides necessary background and hypothesis for this retrospective study.
Material and methods: methods are accurate described, with inclusion and exclusion criteria well defined (osteoporosis or osteopenia, trauma level, trauma analysis, hyponatremia), patient cohort adequately reported, in terms of size, age and sex distribution.
Results section match methods section, being also adequately reported, by all correlations found: comparison between low-energy trauma and control group, hyponatremia in low energy trauma group, osteoporosis distribution and hyponatremia and finally, correlation between bone density and natremia.
Discussions starts with papers’ findings, presented in context of previous studies; limitations of this study are stated, but also the strength of this study: QCT analysis for examining bone quality in hyponatremic patients, with obvious clinical relevance.
Conclusions are short, based on manuscript contained data.
Tables are clear, easy to understand.
References are up to date and correctly formatted.

We thank the reviewer for his/her very positive comments. The reviewer pointed out that our study is based on previous knowledge, but it adds significantly new knowledge since it represents indeed the first study of this kind which includes QCT diagnostics. 

We would like to thank the reviewers for their constructive work as well as their very helpful comments and suggestions.  We hope that the corresponding improvements will now allow that our manuscript will now be accepted for publication in Medicina.

Sincerely,

Dr. med. K. B. Jäckle
(on behalf of all authors)

Reviewer 2 Report

The purpose of your study is to examine the association between chronic hyponatremia and spinal bone quality.

Line 19: describe the definition of “chronic” in the method section.

Line 23: rate is not correct. Is this the proportion ?

Table: add the table where you analyze the association between hyponatremia and osteoporosis.

Line 256: the sentence is wrong because your results is the association, not causal inference.

Line 334:risk factor: Can you conduct multivariable regression analysis due to adjust confounders.

Line 336: increased risk of falls: is it based on the your results?

In conclusion, Please summary of your study shortly.

Author Response

Medicina-1442547
Resubmission of our revised manuscript “Influence of hyponatremia on spinal bone quality and fractures due to low-energy trauma“ by Jäckle et al.

Dear Editors,
Thank you very much for providing a rapid and thorough reviewing process. One of the reviewers praised our work and did not make any suggestions for improvement. The second reviewer has some positive criticism and valuable suggestions which we address in our point-by-point response below. Overall, we are very thankful for the comments and for having the opportunity to resubmit our manuscript, following the advice/suggestions of the reviewer 2 which helped to improve our revised manuscript. 

As you will see, our point-by-point response explains how we addressed each comment of reviewer 2. The changes made in the revised manuscript are highlighted in green.

Kind regards,

K. Jäckle

Point-by-point response to the reviewer´s critique, comments and suggestions.

Reviewer #2: 
Comments and Suggestions for Authors
The purpose of your study is to examine the association between chronic hyponatremia and spinal bone quality.

Line 19: describe the definition of “chronic” in the method section.

We thank the reviewer for pointing this omission. We have added this important information in the methods section. It reads as follows “The serum sodium level on hospital admission of the patient was the major determining factor for the classification of hyponatremia. In our study, chronic hyponatremia was defined as a serum sodium level below 135 mmol/l on admission, and the level had to remain persistently low during the inpatient stay.“ (page 2, line 80-84).

Line 23: rate is not correct. Is this the proportion?

The reviewer is absolutely correct, we have replaced the term “rate” by “proportion”. (page 1, line 23).

Table: add the table where you analyze the association between hyponatremia and osteoporosis.

This table (table 3) is placed in the result section, right below the description of the analysis  of the association of hyponatremia, osteoporosis, bone mineral density and low-energy fractures. In our view, this is the right place for the table, because the reader can read the text and see the table at the same time. However, if the reviewer insists that we add the table somewhere else, we would like to have his/her suggestion for a new position of the table within the manuscript.

Line 256: the sentence is wrong because your results is the association, not causal inference.

We agree with the reviewer and have deleted this sentence. Furthermore, to avoid any kind of misunderstanding, we have added a sentence “….hyponatremia patients, establishing a correlation between hyponatremia and low-energy fractures“ to avoid the impression that there is direct causality.“(page 6; line 257-259). 

Line 334: risk factor: Can you conduct multivariable regression analysis due to adjust confounders.

Based on this suggestion, we consulted the experts in our statistics department. According to their advice, a multivariable regression analysis would not be useful or even misleading since the case numbers are too small. Multivariable regression analysis should only be applied for large-scale longitudinal studies or for studies where a normal distribution is available. None of these requirements exist in our study. Therefore, we would like to follow the advice of our specialists and refrain from this type of analysis.

Line 336: increased risk of falls: is it based on your results?

We thank the reviewer for pointing this out. This was not a result of our study, so we have added one sentence to make this clearer. (page 8, line 338-339).

In conclusion, Please summary of your study shortly.

We have added a short summary in the conclusions section (page 8, line 379-381). It reads: “Taken together, our study suggests that chronic hyponatremia impairs bone quality and that chronic hyponatremia results in an increased prevalence for fractures after low-energy trauma due to the decreased bone quality.“ (page 8, line 344-346).

We would like to thank the reviewers for their constructive work as well as their very helpful comments and suggestions.  We hope that the corresponding improvements will now allow that our manuscript will now be accepted for publication in Medicina.

Sincerely,

Dr. med. K. B. Jäckle
(on behalf of all authors)

Round 2

Reviewer 2 Report

Thank you for the revision.

I think that the level of evidence would be low for a study showing an association without adjusting confounding factors with small sample size.

You should weaken your interpretation based on your results.

Author Response

Medicina-1442547

Resubmission of our revised manuscript “Influence of hyponatremia on spinal bone quality and fractures due to low-energy trauma“ by Jäckle et al.

Dear Editors,

Thank you very much for providing a rapid and thorough reviewing process. The second reviewer has one positive criticism and valuable suggestion which we address in our point-by-point response below. Overall, we are very thankful for the comment and for having the opportunity to resubmit our manuscript, following the advice/suggestion of the reviewer 2 which helped to improve our revised manuscript.

As you will see, our response explains how we addressed the comment of reviewer 2. The change made in the revised manuscript is highlighted in green.

Kind regards,

K. Jäckle

Response to the reviewer´s critique, comment and suggestion.

Reviewer #2:

Thank you for the revision. I think that the level of evidence would be low for a study showing an association without adjusting confounding factors with small sample size. You should weaken your interpretation based on your results.

We agree with the reviewer and have added the following sentence in the discussion section: “This link is mediated via reduced bone quality to the level of osteopenia or even osteoporosis, implying an association rather than causality between fracture risk and hyponatremia. This causion must be noted in the absence of a multivariable regression analysis which cannot be applied in a meaningful manner or would even be misleading, since our study does not involve the required large data sets for such an analysis.” (page 6, line 259-264).

We would like to thank the reviewer again for his/her constructive work as well as his/her very helpful comment and suggestion. We hope that the corresponding improvement will now allow that our manuscript will be accepted for publication in Medicina.

Sincerely,

Dr. med. K. B. Jäckle

(on behalf of all authors)
